# Local Immune Activation and Age Impact on Humoral Immunity in Mice, with a Focus on IgG Sialylation

**DOI:** 10.3390/vaccines12050479

**Published:** 2024-04-29

**Authors:** Priti Gupta, Tibor Sághy, Miriam Bollmann, Tao Jin, Claes Ohlsson, Hans Carlsten, Carmen Corciulo, Cecilia Engdahl

**Affiliations:** 1Department of Rheumatology and Inflammation Research, Sahlgrenska Academy, University of Gothenburg, 413 90 Gothenburg, Sweden; priti.gupta@gu.se (P.G.); tibor.saghy@gu.se (T.S.); miriam.bollmann@gu.se (M.B.); tao.jin@rheuma.gu.se (T.J.); hans.carlsten@rheuma.gu.se (H.C.); 2Department of Internal Medicine and Clinical Nutrition, Sahlgrenska Osteoporosis Centre and Centre for Bone and Arthritis Research, Institute of Medicine, Sahlgrenska Academy, University of Gothenburg, 413 45 Gothenburg, Sweden; claes.ohlsson@medic.gu.se; 3SciLifeLab, University of Gothenburg, 413 90 Gothenburg, Sweden; 4Department of Rheumatology, Sahlgrenska University Hospital, 413 46 Gothenburg, Sweden; 5Department of Pharmacology, Institute of Neuroscience and Physiology, Sahlgrenska Academy, University of Gothenburg, 405 30 Gothenburg, Sweden; carmen.corciulo@gu.se

**Keywords:** humoral immune response, IgG, age, antigen-induced arthritis, IgG-sialylation

## Abstract

Age alters the host’s susceptibility to immune induction. Humoral immunity with circulating antibodies, particularly immunoglobulin G (IgG), plays an essential role in immune response. IgG glycosylation in the fragment crystallizable (Fc) region, including sialylation, is important in regulating the effector function by interacting with Fc gamma receptors (FcγRs). Glycosylation is fundamentally changed with age and inflammatory responses. We aimed to explore the regulation of humoral immunity by comparing responses to antigen-induced immune challenges in young and adult mice using a local antigen-induced arthritis mouse model. This study examines the differences in immune response between healthy and immune-challenged states across these groups. Our initial assessment of the arthritis model indicated that adult mice presented more severe knee swelling than their younger counterparts. In contrast, we found that neither histological assessment, bone mineral density, nor the number of osteoclasts differs. Our data revealed an age-associated but not immune challenge increase in total IgG; the only subtype affected by immune challenge was IgG1 and partially IgG3. Interestingly, the sialylation of IgG2b and IgG3 is affected by age and immune challenges but not stimulated further by immune challenges in adult mice. This suggests a shift in IgG towards a pro-inflammatory and potentially pathogenic state with age and inflammation.

## 1. Introduction

Humoral immunity, an essential component of the adaptive immune system, is characterized by generating antibodies by B- and plasma cells in response to antigens. Immunoglobulin G (IgG) is the most abundant antibody in the circulatory system and facilitates antigen neutralization, complement activation, opsonization for phagocytosis, and antibody-dependent cellular cytotoxicity (ADCC) [1]. The IgG molecule comprises two functional regions: the fragment antigen binding (Fab) region that binds to antigens and the fragment crystallizable (Fc) region that interacts with Fc gamma receptors (FcγRs) on immune cells and mediates the effector functions. In mice, there are three activating receptors, FcγRI, FcγRIII, FcγRIV, and one inhibitory FcγRIIb [2].

Mouse IgG is subclassified into four variants (IgG1, IgG2a/c, IgG2b, and IgG3), each with a different binding affinity with FcγRs and participating at various levels in the complement activation [3]. Despite its relatively low abundance, IgG3 is the most potent immunoglobulin subtype binding to FcγRs due to its extended hinge region, which offers greater flexibility of IgG3 compared to other subtypes [4].

Glycosylation is an enzymatic post-translational modification mediated by glycosyltransferases in the ER-Golgi pathway, in which glycans are covalently attached to peptide molecules. This process is critical for protein folding, stability, and biological function. IgG molecules bear several glycosylation sites in the Fab region, affecting the binding to the antigen, and one conserved site on the CH2 domain in the Fc-region, asparagine at position 297 (Asn-297), involving the interaction with FcγRs [5,6]. This modification features a heptameric complex biantennary glycan structure elongated with five molecules of N-acetylglucosamine, three mannose, two galactose, and two sialic acids at the terminal and one core fucose.

Aging is often defined as a progressive functional decline at multiple levels in an organism, including the physiological system, tissues, cells, and molecules [7,8,9]. Lymphocytes are reduced with age, but immunoglobulins, like IgG, are induced with age with a reduction in IgG glycosylation [8,10,11]. However, the difference between the young and adult impact on humoral immune response and IgG glycosylation patterns remains unexplored in functional models. Our study compares young and adult mice to explore life changes in immune function. This helps us understand how shifts in immune responses in early adulthood can evolve into significant challenges as mice age.

Antigen challenge can trigger an activation of antibody production. This immune induction often results in newly synthesized IgGs with a lower degree of Fc glycosylation aimed at enhancing immune efficacy. Antigen-methylated BSA (mBSA) is used to induce local antigen-induced arthritis in mice. Arthritis is a debilitating inflammatory condition leading to joint damage, including bone erosion and bone loss [12]. Degenerative joint deterioration, as well as the risk of autoimmune diseases, is more prevalent with advancing age.

This study aimed to investigate the interplay between immune activation and the difference between young and adult mice in the humoral immune response by focusing on IgG sialylation. We demonstrated that total IgG, IgG2b, IgG3, and general IgG sialylation (on both Fab+Fc) was age-dependent, whereas IgG1 was influenced by age and immune activation. LC-MS/MS analysis revealed that neither age nor inflammation altered the A-galactosylated forms of IgG2b and IgG3 on the Fc part. However, Fc- sialylation of IgG2b and IgG3 was significantly reduced with both immune challenge and age, suggesting that reduced IgG sialylation may play a vital role in disease response to pathogenesis.

## 2. Material and Methods

### 2.1. Animals and Care

Seven-week-old C57BL/6 (Janvier, Tancom, France) female mice were housed in a standard animal facility at the Laboratory of Experimental Biomedicine at Gothenburg University and fed phytoestrogen-free chow ad libitum (Harlan Rodent Diet, 2016, Stockholm, Sweden).

### 2.2. Ethical Consideration

The study was approved by the ethics committee of the Gothenburg region, Sweden (Engdahl 3020-2020).

### 2.3. Induction of Antigen-Induced Arthritis (AIA) for Local Knee Inflammation

Young growing (2.5 months) (*n* = 6–9) and adult mice (5 months) (*n* = 8–12) were challenged with 1 mg/mL of methylated bovine serum albumin (mBSA) (Sigma Aldrich, Solna, Sweden) dissolved in 1 mL of phosphate-buffered saline (PBS) and emulsified with an equal volume of complete Freund’s adjuvant (Sigma-Aldrich, Solna, Sweden). A total volume of 100 μL was injected intradermally at the base of the tail of the mice (50 μL on each side). After seven days, the mice received a secondary immunization with 5 mg/mL of mBSA in PBS (30 μL) in the right knee joints. A comparison was performed of AIA-induced mice with non-induced naïve mice, both intradermal and local knee injections of PBS. The experimental scheme is shown in Appendix A. Swelling over the arthritic joint was measured daily for the week after the knee injection using a caliper [13]. The differences are plotted in swelling from the day with intra-articular injection, day 0.

### 2.4. Histological Examination

Fourteen days after the primary immune challenge, the mice were anesthetized with ketamine/medetomidine (Pfizer, Strängnäs, Sweden), sacrificed by exsanguination followed by cervical dislocation. Mice’s body weight was measured before termination. The right knees of the mice from both age groups were placed in 4% formaldehyde for 48 h, followed by storage in 70% ethanol and 3X wash in PBS, decalcified in 10% EDTA with 0.1 M tris buffer (pH 6–9.5) for three weeks, dehydrated, and embedded in paraffin. The knees were sectioned in 4 μm and stained with hematoxylin and eosin (H&E) to measure synovial infiltration and bone erosion. Synovitis and bone erosion were blindly graded two times (by researchers CE and PG) using a three-grade histological scoring system (mild: 1, moderate: 2, or severe: 3), and the median value is used for calculation of differences described by Liphardt and colleagues [14]. For osteoclast counts, sections were stained for tartrate-resistant acid phosphatase (TRAP). For TRAP staining, sections were warmed at 60 °C for 30 min and allowed to cool at room temperature. The sections were washed three times in Xylene for ten min each, followed by 100%, 90%, 80%, and 70% ethanol for 2 min each, and finally with water. Sections were incubated for 1.5 h at room temperature in 0.2 M acetate buffer (pH 5), followed by 1.5 h in TRAP buffer, a solution of Naphthol, red-violet, 0.1 M acetate buffer, 0.3 M sodium tartrate, and Triton X-100. After washing the sections with water and staining them for 30 s with counterstain fast green (0.5%), the sections were rinsed in water and 100% ethanol. A glass coverslip was mounted using a Permount mounting medium (Fisher Scientific, Gothenburg, Sweden). The images were acquired using a Zeiss Axioscan Z1 slide scanner (20×) running Zeiss Zen Software 2.3 (Carl Zeis MicroImaging, Jena, Germany). The osteoclast was blindly counted by CE under the supervision of CC. The osteoclast number, area attaching the bone, and the bone surface were determined in the epiphyseal part of the tibia.

### 2.5. Determination of Periarticular Bone Mineral Density (BMD) Using Peripheral Quantitative Computer Tomography (pQCT)

The skin was removed from the hind limb with the knee intact to determine bone loss. Computed tomography analysis was performed with the Stratec peripheral quantitative computed tomography (pQCT) XCT Research M software version 5.4B (Norland, Fort Atkins, WI, USA) at a resolution of 70 μm, as described previously in [15]. Trabecular tibial BMD was determined with epiphysial scanning performed 0.4 mm from the growth plate. The trabecular bone region was defined by setting an inner area to 45% of the total cross-sectioned area. The mid-diaphyseal region of the tibia was used to determine cortical thickness.

### 2.6. Serum Analyses

Blood was collected from the axillary vein in a serum-gel tube (Sarsted, Helsingborg, Sweden) on the day of termination, and the serum was obtained by centrifugation. Antigen-specific (mBSA) anti-IgG was determined with a homemade ELISA as described in [16]. In brief, 96-well plates (Maxisorp, Fisher Scientific, Gothenburg, Sweden) were coated with 0.01 mg/mL mBSA, blocked, and washed with 3% casein (Sigma-Aldrich, Solna, Sweden) before adding the serum (1:100). Bound anti-mBSA antibodies were detected by horseradish peroxidase (HRP)-conjugated rabbit anti-mouse IgG (DAKO, Sigma Aldrich, Solna, Sweden), followed by reaction development with TMB substrate. The absorbance at 450 nm was read by using a microplate reader. Total serum IgM (1:5000); IgGs (1:10,000) (Bethyl Laboratories, Montgomery, TX, USA); and IgG-subtypes IgG1 (1:500), IgG2b (1:5000), and IgG3 (1:10,000) were measured using a commercially available ELISA kit (Thermo Fisher, Invitrogen, Stockholm, Sweden) according to the manufacturer protocol. The sialic acids on the total IgG, including both Fab and Fc parts, were measured with Lectin ELISA, as described previously in [17]. Briefly, an anti-mouse IgG F(ab)2 fragment (Sigma-Aldrich, Solna, Sweden) was used to coat 96-well plates overnight at 4 °C in sodium bicarbonate buffer (0.1 M) followed by blocking using polyvinylpyrrolidone (PVP), serum incubation (1:100), biotinylated Sambucus nigra lectin (SNA) (Vector Laboratories, BioNordika, Solna, Sweden), then streptavidin, and later reaction development.

### 2.7. IgG Purification and Preparation for the Glycoproteomic Analysis

Mouse IgG was purified from the serum samples of both age groups (180–200 μL per mouse) using a commercially available IgG purification kit (Protein G High-Performance Spintrap™ Sigma-Aldrich, Solna, Sweden), according to the manufacturer’s instructions. After the purification of IgG, the concentration of IgG and protein was measured with a commercially available IgG ELISA kit, as used before (Bethyl Laboratories, Montgomery, TX, USA), and a DCA protein kit (Bio-rad, Solna, Sweden), respectively. IgG samples (30 μg protein) were reduced, and Cys residues were derivatized with S-Methyl methanethiosulfonate and cleaved in-solution with trypsin (1:50) in 0.5% sodium deoxycholate using standard procedures. According to the manufacturer’s instructions, the samples were desalted using Pierce C18 desalting columns (Thermo Scientific, Stockholm, Sweden). An Exploris 480 Orbitrap instrument (Thermo Scientific, Stockholm, Sweden) interfaced with an Easy-nLC-1200 chromatography system was used for the nano-LC-MS/MS analysis. The trapping column was an Acclaim Pepmap 100 C18 (Thermo Scientific, Stockholm, Sweden), and the analytical column was in-house packed with Reprosil-Pur (75 μm × 350 mm, particle size 3 μm, Dr Maisch). The flow was set to 300 nL/min, and the 90 min elution gradient was 5% B in A to 45% B. In A over 78 min, then to 100% B over 2 min, and then kept at 100% B for 10 min. A was 0.2% formic acid in the water; B was 80% acetonitrile and 0.2% formic acid in the water. Each sample (2 mL) was analyzed three times consecutively.

The precursor ion MS1 spectra were collected at *m*/*z* 600–2200 at a mass resolution of 120,000. The most intense ions were subjected to MS/MS (MS2), using an isolation window of 3 *m*/*z* units, with higher-energy collision dissociation (HCD) at a normalized collision energy (NCE) of 20 for the first injection, NCE 30 for the second injection, and NCE 38 for the third injection. The MS2 spectra were collected with a resolution of 30,000 with a first mass of *m*/*z* 110. The cycle time was 2 s, and an exclusion time of 12 s was used.

### 2.8. Glycoproteomic Data Analysis

The LC-MS/MS identification of trypsin-digested glycopeptides was made with the Byonic node (Protein Metrics) of Proteome Discoverer (v 2.4, Thermo Scientific), and the precursor ion intensities were extracted with the Minora Feature Detector node. Search settings included the following: the database was Swiss-Prot mouse (17,155 sequences); cleavage after Arg and Lys; two missed cleavages allowed; mass accuracy set to 10 ppm and 30 ppm at MS1 and MS2, respectively; fixed modification was a methyl thio group at Cys; allowed modification was Met oxidation (set to rare 1). The N-glycan database (set to standard 1) was selected to contain 123 complex type, oligomannose, and hybrid structures and included ones typically observed for immunoglobulins, and had 0–3 of either sialic acid Neu5Ac or Neu5Gc glycoforms, 0–1 fucose and 0–1 bisecting GlcNAc. The glycopeptide identities for each sample were filtered to include at least one identity from the three injections, which had a Byonic score >300. Glycopeptide hits were manually verified to contain the expected peptide+GlcNAc ion (Y1 ion), and if Fuc was suggested, the peptide + GlcNAc + Fuc ion (core fucosylation) or an ion at *m*/*z* 512 (antenna fucosylation) had to be present. Further, identities containing Neu5Gc had to include the ions at *m*/*z* 290 and *m*/*z* 308; correspondingly, *m*/*z* 274 and *m*/*z* 292 had to be present for Neu5Ac.

### 2.9. Relative Quantitation of Glycopeptides

The peak intensity ratio (in percentage) for each glycoform was calculated by dividing the Minora-detected precursor ion intensity by the sum of the intensities of the identified glycoforms that shared the same peptide and was presented as average values for the two injections of IgG2b and three injections for IgG3.

### 2.10. Statistical Analyses

Statistical analyses were performed using GraphPad Prism version 9 (GraphPad Software, La Jolla, San Diego, CA, USA). Grubb’s test detected outliers and consequently excluded them from analysis. The Mann–Whitney test was used for the histological assessment. Statistical evaluations were performed using a two-way analysis of variance (2-way ANOVA) to evaluate the effect of age, the effect of immune challenge, and the interaction between age and immune challenge, followed by Fisher’s LSD multiple comparisons post hoc test. The specific glycoforms were analyzed using Fisher’s LSD multiple comparisons. Data are presented as bar graphs and standard error mean (SEM). A *p*-value < 0.05 was considered statistically significant.

## 3. Results

### 3.1. Age Affects Knee Swelling but Not Bone Alteration in mBSA-Induced Arthritis Mouse Model

Body weight depends on both immune challenge and age, and there is no significant interaction between these factors (Figure 1A). The adult mice had a higher body weight than their younger counterparts, while the immune challenge led to weight loss in adults. As expected, a marked increase in knee swelling was observed in mBSA-induced arthritic mice compared to PBS-treated mice (Figure 1B). The adult mice had a further increase in knee swelling compared to their younger counterparts. Arthritic mBSA-treated knee showed increased synovitis and bone erosion compared to the PBS control group, with no age-related differences observed (Figure 1C). Further, trabecular bone mineral density was reduced following mBSA induction in both young and adult mice, with no age-related difference. The cortical bone thickness was increased by age and reduced by immune challenge with no significant interaction effect (Figure 1D). Osteoclast number was assessed on tibial epiphyseal trabecular bone sections. We found a significant increase in the number of osteoclasts in arthritic mice, as measured by the number of osteoclasts per area, the ratio of osteoclasts to the bone perimeter, and the proportion of osteoclast surface to bone surface across both age groups (Figure 1E).

### 3.2. Total IgG, IgG2b, and IgG3, as Well as Sialic Acid on Total IgG, Were Influenced by Age While Immune Challenges Altered IgG1 Levels

We observed no significant interaction between age and immune challenge on the level of immunoglobulin (Figure 2). Following mBSA injection, anti-mBSA specific IgG was detectable in both young and adult AIA immune challenge mice, with no age-dependent difference (Figure 2A). Total IgG concentrations remained unaffected by the immune challenge (Figure 2B). In addition, an age-dependent difference was displayed in both mBSA- and PBS-challenged adult mice, with higher IgG levels in adults compared to their younger counterparts. Different IgG subtypes showed that IgG2b and IgG3 levels increased with age, independent of immune challenges (Figure 2C). IgG1 levels were altered depending on immune challenges, while with ANOVA, an age-dependent increase was also displayed. Similarly, IgM levels increased with age and only a limited response to immune challenges, as indicated in the ANOVA (Appendix A). In terms of sialylation, total IgG (both Fab and Fc parts) was significantly higher in adult mice unaffected by immune activation (Figure 2D).

### 3.3. Both Age and Inflammatory Processes Regulate the Sialylation Patterns of IgG2b and IgG3

Detailed examination of IgG-Fc glycosylation identified specific glycoforms for the IgG2b and IgG3 variants. We identified 13 different glycoforms in the IgG-Fc region of the IgG2b isotype (Uniprot entry P01867, peptide sequence EDYNSTIR) and further grouped them as A-galactosylated (G0), galactosylated (G1/G2), and sialylated (G1S/G2S) (Appendix A). Although no significant interaction between age and immune challenge on IgG2b–glycosylation was found, immune challenge significantly affected the galactosylation and sialylation levels (Figure 3). Inflammation and age did not affect the A-galactosylated IgG2b levels (Figure 3A). Young mBSA-treated mice exhibited significantly higher galactosylation than their PBS counterparts (Figure 3B). Specifically, the highly galactosylated H5N4FG2 isoform with fucose was found to be increased in young mice following the immune challenge (Appendix A). Sialylation of IgG2b was significantly decreased in young mice upon immune challenge, yet no change was observed dependent on the immune induction in the adult mice (Figure 3C). Despite no age-dependent difference in the ANOVA, the post hoc test displayed a significantly lowered frequency of sialylation in the adult PBS control mice compared to young PBS controls. The highly sialylated H5N4FG2 glycoform was decreased in both immune-induced and aged mice groups (Appendix A). The reduction in sialyation was pronounced in adult mice, implying that H5N4FG2 may have a role in modulating the pathogenicity of IgG2b.

In the IgG-Fc region of IgG3 isotype (Uniprot entry P03897, peptide sequence EAQYNSTFR), we identified nine glycoforms and sorted them into A-galactosylated, galactosylated, and sialylated groups (Appendix A). We observed a significant interaction between age and immune challenge for IgG3 galactosylation and sialylation. In comparison, the A-galactosylation and sialylation modulation showed age dependency rather than immune challenge (Figure 4). Similar to IgG2b, immune induction did not alter the A-galactosylated (G0) IgG3 frequency (Figure 4A). However, A-galactosylation was significantly increased in adult immune-challenged mice compared to their young counterpart. This induction was observed only for the H3N4F glycoform (Appendix A). The galactosylation in young immune-induced mice was significantly increased compared to young PBS control (Figure 4B). In immunized mice, we observed a reduction in galactosylated IgG3 in adult mice compared to young mice, but no alteration was observed in non-immune-challenged mice. This was due to a decrease in the highly galactosylated glycoform H5N4F in adult immune-challenged mice compared to young immune-challenged mice and adult PBS (Appendix A). Finally, IgG3 sialylation was significantly reduced in young immune-challenged mice compared to young PBS control mice (Figure 4C). The same significant reduction was demonstrated in adult PBS mice compared to young PBS mice. A decrease in the sialylation of IgG3 was observed in H4N4FG1 and H5N4FG2 in immune-activated and adult mice (Appendix A). The decline in sialylation could be related to the initial decrease in the adult group, which reached an almost saturated condition, thereby restricting further reduction in the immune-activated.

## 4. Discussion

It is well known that adaptive immunity, especially the humoral immune response, diminishes with age, resulting in greater susceptibility to local and systemic immune challenges [18]. This decline reduces responsiveness to treatment and immunization [19]. The inflammation tends to be severe in older individuals, though it remains unclear if there also is a difference between young and adulthood.

We used a well-established antigen-induced model in young and adult mice that involved a systemic challenge, an intra-articular injection of mBSA, resulting in the development of a local mono-arthritis [20]. First, we characterized the arthritis model and showed more pronounced knee swelling in adults compared to their younger counterparts. However, upon examining the arthritic joint, neither synovitis nor bone erosion responded differently in young mice compared to adult mice. This suggests that age could contribute to an increase in inflammatory mediators affecting fluid accumulation in the inflamed joints. The immune challenge led to a significant reduction in trabecular bone BMD and an increased number of osteoclast at the epiphyseal part of the tibia, aligning with our previous findings [15], suggesting that mono-arthritis induces periarticular trabecular bone loss both in young and adult mice. The trabecular BMD did not differ between ages, consistent with prior murine studies [21].

Age-related dysregulation and a swift decline in the immune response have been associated with increased susceptibility to infections and weaker vaccine response [7,22,23]. Pathogen infection triggers an antigen-specific antibody response, as was shown in the current study in mice, where the antigen mBSA-specific IgG increased after the mBSA challenge. It has been suggested that antigen-specific antibody response is predominantly decreased in older people [24]. However, consistent with our previous study in systemic bacteremia, where age did not alter the antigen-specific IgG [25], we show no differences between young and adult mice in mBSA-specific IgG.

In line with previous studies of Staphylococcus infection in mice [25] and elderly humans [26], we found an increase in the older mice in all types of circulating IgG and subtypes. Serum IgG subclass levels were shown to vary with inflammation and age, although it is not constant, as described in several studies [25,27,28]. In this study, IgG1 levels mainly increased following the immune challenge, like our previous study on the Staphylococcus challenge in mice [25]. IgG1 is vital for promoting opsonization and may have a remarkable ability to interact with FcγRs and activate the immune system.

Furthermore, we found that the IgG2b and IgG3 levels increased mainly with age, showing the same pattern as total IgG. This IgG2 finding aligns with a previous study in humans [29]. They found that adults and older human individuals mainly responded with IgG2 antibodies against pneumococcal polysaccharides, regardless of vaccination. The age-dependent alteration of IgG2b coincided with our previous results in Staphylococcus-induced mice [25]. Despite being the least abundant IgG subtype in mouse serum, IgG3 increases with infections [30], especially with protein antigens [31]. Surprisingly, we detected glycoforms of IgG3 with LC-MS/mass spectrometry analysis and confirmed the presence of IgG3 using ELISA in serum, repeated twice with different dilutions. IgG3 has structural flexibility in the hinge region compared to other IgG isotypes, which makes it a potent antibody and highly efficient in mediating Fc-mediated immune functions [32]. A previous study in mice immunized with Legionella pneumophila demonstrated a significant decrease in the serum IgG3 response [33]. In the present study, we found that age, rather than immunological induction, predominantly increases IgG3 levels. Additional studies have suggested that IgG3, unlike IgG1, IgG2a, and IgG2b, but similar to IgM, enhances antibody responses through complement activation [34].

It is well-established that IgG glycosylation, especially terminal sialylation, plays a critical role in inflammation, disease, and aging. The total IgG glycosylation, including Fab and Fc domains, modulates the IgG inflammatory potential and the antigen binding affinity. While some population studies argue that IgG sialylation remains unchanged with age [10,11], others propose a decline [5]. Echoing our prior work with Staphylococcus-challenged mice [25], this study found that sialic acid levels on total IgG increase with age. This elevation correlates with the sialylation of the Fab portion of IgG, which exhibits higher sialylation compared to the Fc portion. Furthermore, total IgG sialylation was assessed using Lectin ELISA, a method that detects sialic acid present on both the Fab and Fc parts.

Further, we investigated the IgG-Fc N-linked glycosylation, which is crucial for modulating IgG effector functions and regulating its inflammatory responses. This glycosylation can enhance pro-inflammatory effects such as activation of the complement system, opsonization, and ADCC. Conversely, it also supports anti-inflammatory actions through mechanisms like inhibition of complement activation; increased sialylation, which promotes engagement with inhibitory receptors leading to reduced inflammation; and interaction with FcγRIIB to suppress immune cell activation. It is commonly understood that inflammation and aging showed an increase in the A-galactosylated (G0) form of IgG, thereby enhancing the capacity of IgG to bind with FcγRs. This has been described in patients with immune induction [5,35,36]. Some functional studies in murine autoimmunity models showed an increase in A-galactosylation of IgG [37,38]. Our study showed only an induction in the A-galactosylated IgG3 in adult immune-challenged mice. This phenomenon was explicitly observed in the highly expressed H3N4F glycoform.

Similarly, decreased levels of galactosylation and sialylation in population studies, following the immune induction [13] and aging [39,40,41], promote IgG binding and activation. In this murine study, we observed that the sialylation of IgG2b and IgG3 decreased with age, young compared to adult, and inflammation. The decline in sialylation for both IgG isotypes was associated with the reduction in highly sialylated G2S. Previous studies have shown that G2S is crucial in inhibiting ADCC, reducing its binding affinity with FcγRs, and promoting a shift toward an anti-inflammatory response of IgG [42]. This detailed investigation of IgG-Fc sialylation presents a counterpoint to IgG’s overall sialic acid content. Previous studies that reported a decrease in IgG galactosylation under inflammatory conditions and with increasing age were reviewed by Gudelj et al. [5]. In contrast, our findings demonstrated that young mice exhibit an increase in galactosylation upon immune challenge. This discrepancy may be attributed to immunological stimulation having a greater impact on sialylation in mice, while galactosylation is more commonly associated with humans. Furthermore, this was accompanied by a parallel reduction in IgG-Fc sialylation, a phenomenon previously observed in arthritis patients treated with anti-TNF [43].

Nevertheless, we observed a decline in IgG3-galactosylation in adult immune-challenged mice compared to their younger counterparts. This decrease was attributed to the highly galactosylated H5N4F structure reduction, linked to the pro-inflammatory response of IgG3 [5]. It is important to highlight that IgG-Fc is commonly only galactosylated in humans, whereas IgG-Fc is frequently sialylated in mice. These distinct glycosylation patterns could maintain pro-inflammatory and anti-inflammatory responses. Our results indicate that the humeral immune response does not only change in old individuals; here, we also display an alteration between young and adult mice. This is, to our knowledge, the first time demonstrating that IgG-Fc sialylation is altered in adult mice compared to their younger counterparts, and this, as well as IgG-Fc galactosylation, needs to be further investigated in humans.

## 5. Conclusions

In conclusion, both immune challenges and age, young compared to adult, significantly affect humoral immunity and IgG-Fc glycosylation. In this study, mice age increased IgG levels and decreased the IgG-Fc sialylation, resulting in higher levels of antibodies with a higher inductive capacity. Interestingly, in the adult mice, the immune challenge could not further increase the activation capability, decrease the IgG-Fc sialylation, and hence play a role in humoral protection. Exploring therapeutic strategies and targeting IgG glycosylation patterns at subclass-specific levels could improve treatment efficacy and reduce adverse effects in young adults.

## Figures and Tables

**Figure 1 vaccines-12-00479-f001:**
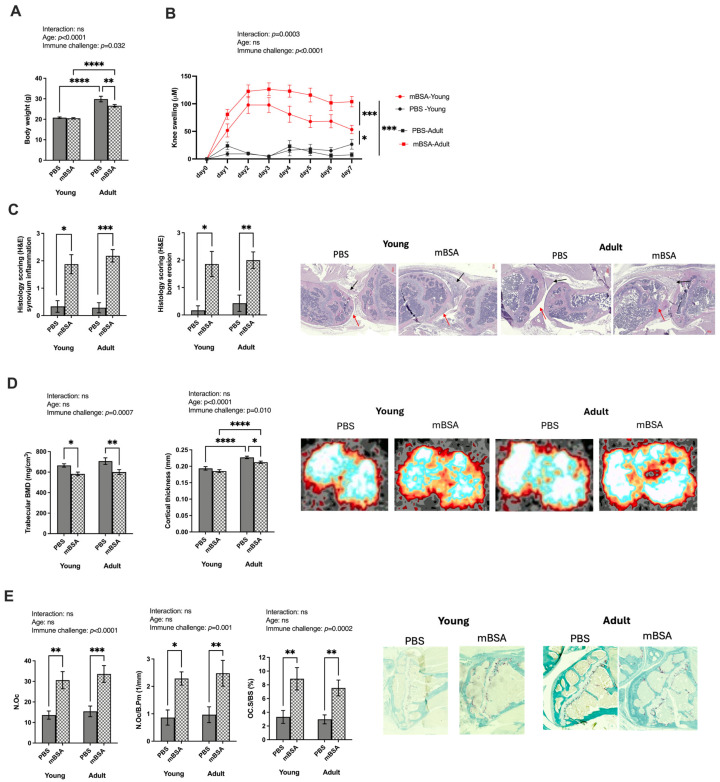
Age affects knee swelling but not bone alteration in mBSA-induced arthritis mouse model. AIA and control young and adult mice groups were immunized with mBSA and PBS (PBS young = 6, mBSA young = 9, PBS adult = 8, and mBSA adult = 12). Antigen challenge was repeated intra-articularly after seven days, and knee joint swelling was measured daily over seven days: (**A**) body weight of the mice at termination; (**B**) the difference in knee swelling from baseline in micrometers (µm) in mice challenged with mBSA of both young and adult groups; (**C**) histological scoring (0–3) of synovial inflammation and bone erosions after seven days from the difference from the baseline. On the right are representative images of hematoxylin and eosin staining of both young and adult PBS vs. mBSA knee joints. The red arrow indicates bone erosion, and the black arrow indicates synovial inflammation. Bone phenotype assessment with pQCT analysis: (**D**) trabecular bone BMD in mg/cm^3^; cortical thickness on the diaphyseal region (mm); representative images of pQCT analysis both in young and adult PBS vs. mBSA tibia; (**E**) number of osteoclast on the tibial epiphyseal region; the N.Oc/B.Pm (1/mm) and Ob.S/BS. Statistical evaluations were performed using a two-way analysis of variance (2 way ANOVA) to evaluate the effect of age, the effect of immune challenge, and the interaction between age and immune challenge. Two-way analysis to compare the interaction is stated at the top of each figure, followed by Fisher’s LSD multiple comparisons test where a significant difference is indicated with stars in the figure. Results are shown as bar graphs with mean ± SE. For histological examination, Mann–Whitney non-parametric tests were performed. * *p* < 0.05, ** *p* < 0.01, *** *p* < 0.001 and **** *p* < 0.0001. AIA—antigen induced arthrthis; mBSA—methylated bovine serum albumin; BMD—bone mineral density; B.Pm—bone perimeter; BS—bone surface.

**Figure 2 vaccines-12-00479-f002:**
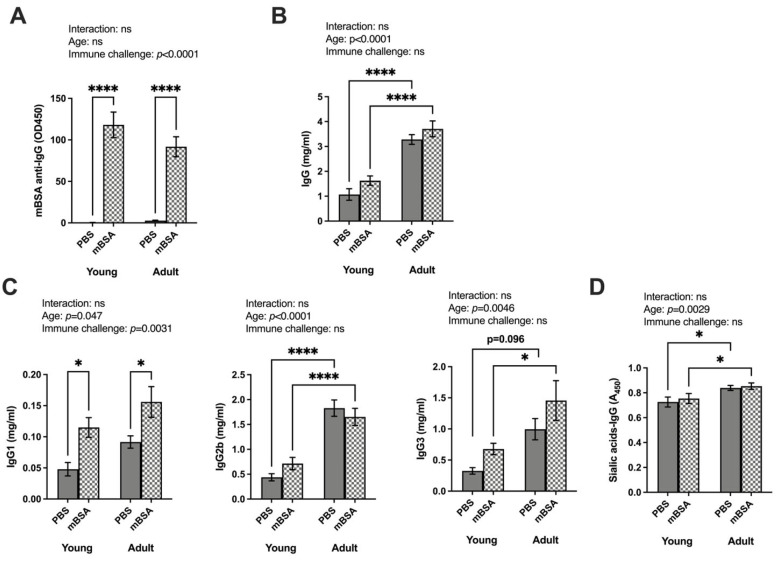
Total IgG and all subtypes, as well as sialic acid on total IgG, were influenced by age, while both IgG1 also responded to immune challenges. Both AIA and control young and adult mice groups were immunized with mBSA and PBS (PBS young = 6, mBSA young = 9, PBS adult = 8, and mBSA adult = 12). Serum was collected at termination on day 14 from the primary systemic immunization: (**A**) mBSA-specific anti-IgG (OD450); (**B**) total IgG (mg/mL); (**C**) IgG subclass: IgG1 (mg/mL), IgG2b (mg/mL), and IgG3 (mg/mL); (**D**) total sialic acid on total IgG (A450). Statistical evaluations were performed using a two-way analysis of variance (2 way ANOVA) to evaluate the effect of age, the effect of immune challenge, and the interaction between age and immune challenge. Two-way analysis to compare the interaction is stated at the top of each figure, followed by Fisher’s LSD multiple comparisons test where a significant difference is indicated with stars in the figure. Results are shown as bar graphs with mean ± SE. * *p* < 0.05, and **** *p* < 0.00001. AIA—antigen induced arthrthis; mBSA—methylated bovine serum albumin.

**Figure 3 vaccines-12-00479-f003:**
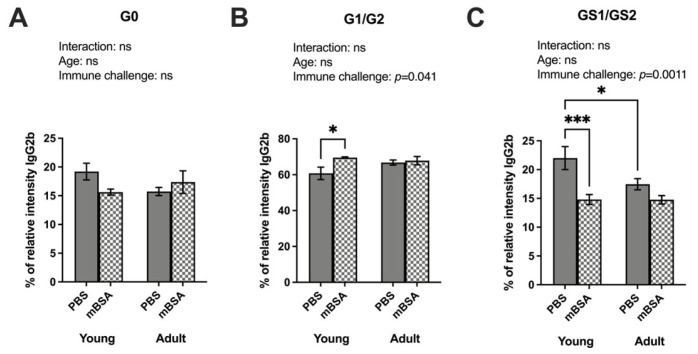
Both age and inflammatory processes regulate the sialylation patterns of IgG2b. Both AIA and control young and adult mice groups were immunized with mBSA and PBS (PBS young = 6, mBSA young = 9, PBS adult = 8, and mBSA adult = 12). IgG was purified from mice serum. Mass spectrometry-based analysis of sialic acids and galactose on IgG subtypes; IgG2b (peptide sequence EDYNSTIR) were measured at termination: (**A**) sum of relative percentage of G0 form as graphically displayed above on IgG2b (% of total IgG2b); (**B**) sum of relative percentage of total galactose G1/G2 form as graphically displayed above on IgG2b (%); (**C**) sum of relative percentage of total sialic acid GS1/GS2 form as graphically displayed above on IgG2b (%). Two-way analysis to compare the interaction is stated at the top of each figure, followed by Fisher’s LSD multiple comparisons test where a significant difference is indicated with stars in the figure. Results are shown as bar graphs with mean ± SE. * *p* < 0.05, and *** *p* < 0.0001. AIA—antigen induced arthrthis; mBSA—methylated bovine serum albumin. G0: A-galactosylated; G1: mono-galactosylated; G2: di-galactosylated; G2S1: mono-sialylated; and G2S2: di-sialylated glycoforms.

**Figure 4 vaccines-12-00479-f004:**
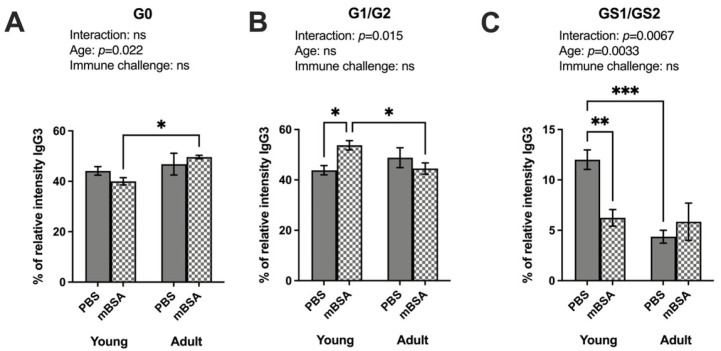
Both age and inflammatory processes regulate the sialylation patterns of IgG3. Both AIA and PBS control young and adult mice groups were immunized with mBSA and PBS (PBS young = 6, mBSA young = 9, PBS adult = 8, and mBSA adult = 12). IgG was purified from mice serum. Mass spectrometry-based analysis of sialic acids and galactose on IgG subtypes, IgG3 (peptide sequence EAQYNSTFR): (**A**) sum of relative percentage of G0 form as graphically displayed above on IgG3 (%); (**B**) sum of relative percentage of total galactose G1/G2 form as graphically displayed above on 3 (%); (**C**) sum of relative percentage of total sialic acid GS1/GS2 form as graphically displayed above on IgG3 (%). Two-way analysis to compare the interaction is stated at the top of each figure, followed by Fisher’s LSD multiple comparisons test where a significant difference is indicated with stars in the figure. Results are shown as bar graphs with mean ± SE. * *p* < 0.05,** *p* < 0.01 and *** *p* < 0.0001. AIA—antigen induced arthrthis; mBSA—methylated bovine serum albumin. G0: A-galactosylated; G1: mono-galactosylated; G2: di-galactosylated; G2S1: mono-sialylated; G2S2: di-sialylated glycoforms.

## Data Availability

The data that support the findings of this study are deposited in the Figshare repository https://doi.org/10.6084/m9.figshare.25452409.

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
