# Peer review of "Local Immune Activation and Age Impact on Humoral Immunity in Mice, with a Focus on IgG Sialylation"

_vaccines, 2024, doi:10.3390/vaccines12050479_

Round 1

Reviewer 1 Report

Comments and Suggestions for Authors

Dear Editor and Authors,

The article “Local Immune Activation and Age Impact on Humoral Immunity in Mice, with a Focus on IgG Sialylation” describes nicely the impact of age (young adult vs adult mice)  on the induction of local inflammation (immune activation)  and on the systemic humoral immune response with focus on sialylation, but also the effect of age on different IgG levels and antigen specific antibodies. The article is clearly and well written and provides interesting information about the influence of age on the humoral, but also local immune response. The M&M and result/discussion section could be extended with additional information, especially about the local immune response, which will improve the publication for a broader audience.

Suggestions for authors:

Line 51-59 : This paragraph describes the relevance of glycosylation. However no references were added. Could the authors add some references, this will also help readers who are less familiar with glycosylation

Line 60: Aging. Could the authors make the aging in general more clear. Is this the aging after the adulthood, or is this the aging of the young  immature animal to adult animal. This are two different things. Perhaps this can also be added to the conclusion. What is the influence of age, is this positive or negative, how should we interpret the data. Young animals have in general less IgG and a higher sialylation of IgG2b and IgG3 compared to adult animals, what is the influence in disease response and pathogenesis of this finding for young animals

Line 92-102 :  1) please add group size in text and Figure S1; 2) add weight measurement; 3) describe the measuring of arthritic joint more specific. Add reference?

Line 104-122: Histological examination

1)      Researchers CE and PG both evaluated the histology of the joints. Did both researchers evaluate all joints and did they reach consensus for differences in grading?. Authors could consider to add the results of the histological scoring. Could the authors be more specific about qualifications or researchers, are they boarded veterinary pathologists?

2)      TRAP evaluation. Could authors be more precise about this. How many microscopic fields were counted, did they use software for the counting etc.

Line 220:  Perhaps the authors could add a supplemental figure with the TRAP stain to show the readers the osteoclasts which were stained

Line 214:  Would it be possible to characterize the synovitis (type of inflammatory cell, oedema, fibrin, necrosis etc).  Was there a difference in morphology between young and adult mice? Why was the knee swelling more pronounced in the adult? This should be visible in histology. Perhaps something to add to discussion.

Line 214-215: “indicating an age-related amplification influenced by the interaction between age 214 and immune challenge.” Perhaps more appropriate for discussion. Chance to discuss the difference in local immune response for young adult and adult mice, which causes the difference in knee swelling.

Line 349: Could the authors elaborate more on the selected age of the mouse model. There is only a difference of 2,5 months. This is sufficient according to literature. For instance, why did the authors not select adult mice and old mice?

General discussion: the authors could consider to translate their results to human medicine, which I think is the ultimate goal of this study. What does this indicate for vaccination/disease susceptibility of young adults compared to adults? Could these results also be translated to old/aging people?

Author Response

Subject: Response to reviewer comments 

Manuscript ID: vaccines-2953621

Dear reviewer 1

Thank you for your thorough investigation of our manuscript. This greatly helps us improve our manuscript, and we hope that you feel that we have answered all your questions. Below, we write a point-by-point answer to all your suggestions.

Suggestions for authors: Line 51-59 : This paragraph describes the relevance of glycosylation. However, no references were added. Could the authors add some references, this will also help readers who are less familiar with glycosylation

Response: We appreciate the suggestion and have included the relevant references in the paragraph.

We changed the text in the introduction and included reference Lines 51-59: “Glycosylation is an enzymatic post-translational modification mediated by glycosyltransferases in the ER-Golgi pathway, in which glycans are covalently attached to peptide molecules. This process is critical for protein folding, stability, and biological function. IgG molecules bear several glycosylation sites in the Fab region, affecting the binding to the antigen, and one conserved site on the CH2 domain in the Fc-region, asparagine at position 297 (Asn-297), involving the interaction with FcγRs [5,6]. This modification features a heptameric complex biantennary glycan structure elongated with five molecules of N-acetylglucosamine, three mannose, two galactose, and two sialic acids at the terminal and one core fucose.”

Suggestions for authors: Line 60: Aging. Could the authors make the aging in general more clear. Is this the aging after the adulthood, or is this the aging of the young immature animal to adult animal. This are two different things. Perhaps this can also be added to the conclusion. What is the influence of age, is this positive or negative, how should we interpret the data. Young animals have in general less IgG and a higher sialylation of IgG2b and IgG3 compared to adult animals, what is the influence in disease response and pathogenesis of this finding for young animals

Response: Thank you for pointing out that we need to clarify this aspect. We have rewritten this and hope that our message is now clearer that we have investigated the difference in the humoral immune system between young and adult mice.

We changed the text in the abstract lines 23-28, “We aimed to explore the regulation of humoral immunity by comparing responses to antigen-induced immune challenges in young and adult mice using a local antigen-induced arthritis mouse model. This study examines the differences in immune response between healthy and immune-challenged states across these groups. Our initial assessment of the arthritis model indicated that adult mice presented more severe knee swelling than their younger counterparts.”

In the introduction lines 72-81,” Aging is often defined as a progressive functional decline at multiple levels in an organism, including the physiological system, tissues, cells, and molecules [7-9]. Lymphocytes are reduced with age, but immunoglobulins, like IgG, are induced with age with a reduction of IgG glycosylation [8,10,11]. However, the difference between the young and adult impact on humoral immune response and IgG glycosylation patterns remains unexplored in functional models. Our study compares young and adult mice to explore life changes in immune function. This helps us understand how shifts in immune responses in early adulthood can evolve into significant challenges as mice age.”

We have also rewritten the conclusion to clarify the influence in pathogenesis

In the conclusion lines 607-612, “In conclusion, both immune challenges and age, young compared to adult, significantly affect humoral immunity and IgG-Fc glycosylation. In this study, mice age increased IgG levels and decreased the IgG-Fc sialylation, resulting in higher levels of antibodies with a higher inductive capacity. Interestingly, in the adult mice, the immune challenge could not further increase the activation capability, decrease the IgG-Fc sialylation, and hence play a role in humoral protection,”

Suggestions for authors: Lines 92-102:  1) please add group size in text and Figure S1; 2) add weight measurement; 3) describe the measuring of arthritic joint more specific. Add reference?

Response: We appreciate the reviewer's insightful observation.

1) Group size is included in the manuscript. Now lines 107-131, “Young growing (2.5 months) (n=6-9) and adult mice (5 months) (n=8-12) were challenged with 1 mg/ml of methylated bovine serum albumin (mBSA) (Sigma Aldrich, Sweden) dissolved in 1 ml of phosphate-buffer saline (PBS) and emulsified with an equal volume of complete Freund’s adjuvant (Sigma -Aldrich, Sweden).”

2) We have updated with weight measurement in S1.

3) The arthritic swelling is described more accurately. Now lines 105-107, ” Swelling over the arthritic joint was measured daily for the week after the knee injection using a caliper [13]. The differences are plotted in swelling from the day with intraarticular injection, day 0.”

Suggestions for authors: Line 104-122: Histological examination 1) Researchers CE and PG both evaluated the histology of the joints. Did both researchers evaluate all joints and did they reach consensus for differences in grading?. Authors could consider to add the results of the histological scoring. Could the authors be more specific about qualifications or researchers, are they boarded veterinary pathologists? 2) TRAP evaluation. Could authors be more precise about this. How many microscopic fields were counted, did they use software for the counting etc.

Response: Thank you for pointing this out.

1) Both researchers evaluated the histological scoring blinded two times, the median value was then used in the manuscript. CE has performed this assessment several times and started with great supervision from the veterinary pathologist. PG is taught by CE and additional supervision from the veterinary pathologist. This is now further explained in the MM. Now lines 147-150, ”Synovitis and bone erosion were blindly graded two times (by researchers CE and PG) using a three-grade histological scoring system (mild:1, moderate:2, or severe:3), and the median value is used for calculation of differences described by Liphardt and colleagues [14].”

CE and CC graded the TRAP staining. This is now further explained in the MM, lines 159-163. “The images were acquired using a Zeiss Axioscan Z1 slide scanner (20x) running Zeiss Zen Software (Carl Zeis MicroImaging, Jena, Germany). The osteoclast was blindly counted by CE under the supervision of CC. The osteoclast number, area attaching the bone, and the bone surface were determined in the epiphyseal part of the tibia.”

Suggestions for authors: Line 220:  Perhaps the authors could add a supplemental figure with the TRAP stain to show the readers the osteoclasts which were stained.

Response: Of course, we understand the concept of viewing the osteoclasts in the epiphyseal bone, so we include this in Figure 1E.

Suggestions for authors: Line 214:  Would it be possible to characterize the synovitis (type of inflammatory cell, oedema, fibrin, necrosis etc).  Was there a difference in morphology between young and adult mice? Why was the knee swelling more pronounced in the adult? This should be visible in histology. Perhaps something to add to discussion.

Response: Thanks for this comment. When characterized synovitis, there were no obvious changes between young and adult mice. We were also surprised when no difference was apparent in the synovitis scoring. However, as indicated, the scoring of synovitis is a blunt method. We believe that the swelling is possibly dependent on more fluid around the joint. This aspect was already lifted in the discussion, but we now rephrased it to make it clearer.

Lines 446-454 ”First, we characterized the arthritis model and showed more pronounced knee swelling in adults compared to their younger counterparts. However, upon examining the arthritic joint, neither synovitis nor bone erosion responded differently in young mice compared to adult mice. This suggests that age could contribute to an increase of inflammatory mediators affecting fluid accumulation in the inflamed joints.”

Suggestions for authors: Line 214-215: “indicating an age-related amplification influenced by the interaction between age 214 and immune challenge.” Perhaps more appropriate for discussion. Chance to discuss the difference in local immune response for young adult and adult mice, which causes the difference in knee swelling.

Response: Thanks for pointing this out. This row is now removed from the results and the aspects are discussed. See previous suggestions for authors.

Suggestions for authors: Line 349: Could the authors elaborate more on the selected age of the mouse model. There is only a difference of 2,5 months. This is sufficient according to literature. For instance, why did the authors not select adult mice and old mice?

Response: Thank you for pointing out that we need to clarify this aspect. Our ethical approval constrains us from using older mice, which influenced our decision to limit the age range studied. We have now rewritten this part and hope that our message is clear that we like to investigate the difference in the humoral immune system between young and adults. Line 349 is rewritten now as in line 423-427. “It is well known that adaptive immunity, especially the humoral immune response, diminishes with age, resulting in greater susceptibility to local and systemic immune challenges [18]. This decline reduces responsiveness to treatment and immunization [19]. The inflammation tends to be severe in older individuals, though it remains unclear if there also is a difference between young and adulthood.”

Suggestions for authors: General discussion: the authors could consider to translate their results to human medicine, which I think is the ultimate goal of this study. What does this indicate for vaccination/disease susceptibility of young adults compared to adults? Could these results also be translated to old/aging people?

Response: The reviewer's point is interesting. Of course, we would like to translate our murine results to humans, but there are still differences between the species. We indeed tried to rephrase the discussion and included a small part at the end.

Line 600-605. ”Our results indicate that the humeral immune response changes not only in old individuals, but here, we display an alteration also between young and adult mice. This is, to our knowledge the first time to show that IgG-Fc sialylation is altered in adult mice compared to their younger counterparts, and this, as well as IgG-Fc galactosylation, needs to be further investigated in humans.”

Thank you for considering our work.

Sincerely,

Cecilia Engdahl PhD,

Associate Professor

Sahlgrenska Osteoporosis Centre, Centre for Bone and Arthritis Research

Reviewer 2 Report

Comments and Suggestions for Authors

The manuscript by Gupta, et al. examines the effects of age and immune activation on young and adult mice antibody concentration and IgG sialylation using an arthritis model. This work implicates an age-associated effect in relatively young mice on total serum IgG as well as sialyation status of IgG. In addition, immune activation negatively impacts sialylation in younger mice and increases concentrations of IgG1.  While the experiments conducted are appropriate based on available tools, there are several comments and questions that the authors could address to improve the manuscript for readers.

Comments:

1.        Discussion of ageing and immunosenescence in the introduction seems out of place here as this study does not focus on older mice, instead comparing young and adult mice. This point could be more strongly expressed throughout the paper.

2.        Given C57Bl/6 mice cannot make IgG2a, instead having the equivalent IgG2c, it is unclear whether IgG2a levels measured are truly representative of the IgG2c population and it would be best to measure this population directly. Specifics on which kit/antibodies were used should be included in methods to make more clear what has been measured. 

3.        For Figure 2D, were similar concentrations of total IgG loaded into each well to control for the variance in serum IgG concentrations between the younger and adult mice? These details should be provided in methods which will enable interpretation of relative sialylation levels between samples.

4.        The text and p value given in Figure 2C show an age-related increase in IgG1, but this should also be displayed on the figure itself.

5.        Why does total sialylation increase with age, but sialylation of specific isoforms decrease? Were sialylation levels compared for each subclass between ages/immune challenge states?

6.        In lihe 281, it is stated that inflammation and age affect A-galactosylated IgG2b levels, but there is no significance shown in Figure 3A. 

7.        The sentence starting in line 326 seems incomplete.

8.        In line 348, it is unclear what is meant by “effect.” This sentence could be revised to make more clear the aim of the study.

9.        In the discussion, there is significant comparison of this study to studies on effects of ageing on the immune system. Indeed, one previous study’s “young” population included the same aged mice as the “older” population in this study. Thus, the mice in this study were ostensibly well within the adult range and much younger than what would be considered “older adult” (and as such, immunosenesence is likely not what is occurring here). It would be more relevant in this study to discuss results in terms of effects of immune system maturation, from young adult to adult, on humoral immunity.

10.  In line 374, human IgG1 levels are mentioned in relation to mouse IgG1, but it is not clear based on their functions that human IgG1 is equivalent to mouse IgG1. Similarly, given they are also not equivalent, comparisons between mouse IgG2, IgG3 and human IgG2, IgG3 should be modified.

11.  The sentence starting in line 417 seems incomplete.

12.  Lines 389, 398, 416, discussion about how the current study’s data are contradictory to the previous literature is not sufficient. More details about how it contradicts and potential reason why should be given.

Author Response

Subject: Response to reviewer comments 

Manuscript ID: vaccines-2953621

Dear reviewer 2

Thank you for the opportunity to revise our manuscript and for the constructive comments from the reviewers. We have carefully considered each comment and have made corresponding revisions to the manuscript. Below, we provide a point-by-point response to the comments raised.

Comment: Discussion of ageing and immunosenescence in the introduction seems out of place here as this study does not focus on older mice, instead comparing young and adult mice. This point could be more strongly expressed throughout the paper.

Response: We appreciate the reviewer's insightful observation. In response to your comment, we refined the text to better align with the scope of our study. Instead of emphasizing aging and immunosenescence broadly, we will adjust our focus to discuss the developmental changes in the immune system as mice transition from young to adult stages.

We changed the text in the abstract lines 23-28, “We aimed to explore the regulation of humoral immunity by comparing responses to antigen-induced immune challenges in young and adult mice using a local antigen-induced arthritis mouse model. This study examines the differences in immune response between healthy and immune-challenged states across these groups. Our initial assessment of the arthritis model indicated that adult mice presented more severe knee swelling than their younger counterparts.”

In the introduction lines 72-81,” Aging is often defined as a progressive functional decline at multiple levels in an organism, including the physiological system, tissues, cells, and molecules [7-9]. Lymphocytes are reduced with age, but immunoglobulins, like IgG, are induced with age with a reduction of IgG glycosylation [8,10,11]. However, the difference between the young and adult impact on humoral immune response and IgG glycosylation patterns remains unexplored in functional models. Our study compares young and adult mice to explore life changes in immune function. This helps us understand how shifts in immune responses in early adulthood can evolve into significant challenges as mice age.”

In discussion Lines 423-427, ”It is well known that adaptive immunity, especially the humoral immune response, diminishes with age, resulting in greater susceptibility to local and systemic immune challenges [18]. This decline reduces responsiveness to treatment and immunization [19]. The inflammation tends to be severe in older individuals, though it remains unclear if there also is a difference between young and adulthood.”

Line 456-463, “Age-related dysregulation and a swift decline in the immune response have been associated with increased susceptibility to infections and weaker vaccine response [7,22,23]. Pathogen infection triggers an antigen-specific antibody response, as was shown in the current study in mice, where the antigen mBSA-specific IgG increased after the mBSA challenge. It has been suggested that antigen-specific antibody response is predominantly decreased in older people [24]. However, consistent with our previous study in systemic bacteremia where age did not alter the antigen-specific IgG [25], we show no differences between young and adult mice in mBSA-specific IgG.”

We have also rewritten the conclusion to clarify the influence in pathogenesis

In the conclusion lines 607-612, “In conclusion, both immune challenges and age, young compared to adult, significantly affect humoral immunity and IgG-Fc glycosylation. In this study, mice age increased IgG levels and decreased the IgG-Fc sialylation, resulting in higher levels of antibodies with a higher inductive capacity. Interestingly, in the adult mice, the immune challenge could not further increase the activation capability, decrease the IgG-Fc sialylation, and hence play a role in humoral protection,”

Comment: Given C57Bl/6 mice cannot make IgG2a, instead having the equivalent IgG2c, it is unclear whether IgG2a levels measured are truly representative of the IgG2c population and it would be best to measure this population directly. Specifics on which kit/antibodies were used should be included in methods to make more clear what has been measured.

Response: Thank you for highlighting this aspect regarding the IgG2a/IgG2c distinction in C57Bl/6 mice. We were previously unaware of the different isoforms in C57Bl/6 mice as this kit has been used to measure the IgG2a response in C57BL6 mice by Youdi et al.; 2020 (https://www.frontiersin.org/journals/immunology/articles/10.3389/fimmu.2020.00913/full). However, we have carefully revised the relevant sections of our manuscript and decided to remove the section regarding the IgG2a to ensure clarity.

We, therefore, rewrote the manuscript and excluded the IgG2a result in introduction line 91-93, “We demonstrated that total IgG, IgG2b, IgG3, and general IgG sialylation (on both Fab+Fc) was age-dependent, whereas IgG1 was influenced by age and immune activation.”

In material and method, lines 191-194, “Total serum IgM (1:5000), IgGs (1:10,000) (Bethyl Laboratories), and IgG-subtypes IgG1 (1:500), IgG2b (1:5000), and IgG3 (1:10,000) were measured using a commercially available ELISA kit (Thermo Fisher, Invitrogen, Sweden) according to the manufacturer protocol.“

In the result section, lines 301-302, “3.2. Total IgG, IgG2b, and IgG3 as well as sialic acid on total IgG, were influenced by age while immune challenges altered IgG1 levels”

And in the discussion, lines 471-477, “Furthermore, we found that the IgG2b and IgG3 levels increased mainly with age, showing the same pattern as total IgG. This IgG2 finding aligns with a previous study in human [29]. They found that adults and older human individuals mainly responded with IgG2 antibodies against pneumococcal polysaccharides, regardless of vaccination. The age-dependent alteration of IgG2b coincided with our previous results in staphylococcus-induced mice.”

Comment: For Figure 2D, were similar concentrations of total IgG loaded into each well to control for the variance in serum IgG concentrations between the younger and adult mice? These details should be provided in methods which will enable interpretation of relative sialylation levels between samples.

Response: We thank the reviewer for this observation and for your comment regarding Figure 2D. In our initial approach, we diluted serum samples to a 1:10,000 ratio before loading them into each well without specifically adjusting for the variance in serum IgG concentrations between younger and adult mice.

To address this comment, we have updated the relevant section in material and method lines 191-194, “Total serum IgM (1:5000), IgGs (1:10,000) (Bethyl Laboratories), and IgG-subtypes IgG1 (1:500), IgG2b (1:5000), and IgG3 (1:10,000) were measured using a commercially available ELISA kit (Thermo Fisher, Invitrogen, Sweden) according to the manufacturer protocol.“

Comment: The text and p value given in Figure 2C show an age-related increase in IgG1, but this should also be displayed on the figure itself.

Response: Thank you for pointing out that we need to clarify this part. The significant difference indicated by the asterisk (*) in the figure is based on the post hoc analysis, not on Two-Way ANOVA. Using post hoc analysis, we found no age-related changes in IgG1. However, using Two-Way ANOVA, we did identify an age-related difference.

We, therefore, clarify this in all figure legends, “Two-way analysis to compare the interaction, are stated in the top of each figure followed by Fisher’s LSD multiple comparisons test where significant difference is indicated with stars in the figure.”

Comment: Why does total sialylation increase with age, but sialylation of specific isoforms decrease? Were sialylation levels compared for each subclass between ages/immune challenge states?

Response: The reviewer raised an interesting point. When we measured total sialylation using ELISA, we analyzed both the Fab and Fc regions of IgG. Notably, the Fab region contains four glycosylation sites, whereas the Fc region has only one. This discrepancy likely compensated for the results. In contrast, our Mass Spectrometry analysis specifically targeted the IgG-Fc sialylation. Here, we observed changes associated with both age and inflammation, aligning with prior studies. Additionally, sialylation levels varied among IgG subtypes; for instance, IgG2b exhibited higher sialylation compared to IgG3.

Comment: In line 281, it is stated that inflammation and age affect A-galactosylated IgG2b levels, but there is no significance shown in Figure 3A.

Response: Thank you for pointing this out. We have edited the manuscript and revised the statement. Now, in lines 347-350, “Although no significant interaction between age and immune challenge on IgG2b- glycosylation was found, immune challenge significantly affected the galactosylation and sialylation levels (Figure 3). Inflammation and age did not affect the A-galactosylated IgG2b levels (Figure 3A).”

Comment: In line 348, it is unclear what is meant by “effect.” This sentence could be revised to make more clear the aim of the study.

Response: We thank the reviewer for this observation. To clarify, we have expanded our discussion on IgG-Fc N-linked glycosylation’s role in modulating IgG effector functions. Now in lines 566-572, ”Further, we investigated the IgG-Fc N-linked glycosylation, which is crucial for modulating IgG effector functions and regulating its inflammatory responses. This glycosylation can enhance pro-inflammatory effects such as activation of the complement system, opsonization, and ADCC. Conversely, it also supports anti-inflammatory actions through mechanisms like inhibition of complement activation, increased sialylation which promotes engagement with inhibitory receptors leading to reduced inflammation, and interaction with FcγRIIB to suppress immune cell activation.”

Comment: In the discussion, there is significant comparison of this study to studies on effects of ageing on the immune system. Indeed, one previous study’s “young” population included the same aged mice as the “older” population in this study. Thus, the mice in this study were ostensibly well within the adult range and much younger than what would be considered “older adult” (and as such, immunosenesence is likely not what is occurring here). It would be more relevant in this study to discuss results in terms of effects of immune system maturation, from young adult to adult, on humoral immunity.

Response: Our ethical approval constrains us from using older mice, which influenced our decision to limit the age range studied. We acknowledge the importance of your observations and have updated the manuscript. Please look at the changes outlined in the first comment for more details.

Comment: In line 374, human IgG1 levels are mentioned in relation to mouse IgG1, but it is not clear based on their functions that human IgG1 is equivalent to mouse IgG1. Similarly, given they are also not equivalent, comparisons between mouse IgG2, IgG3 and human IgG2, IgG3 should be modified.

Response: While mice IgG1 and human IgG1 belong to the same immunoglobulin subclass (IgG1), they differ in structure, function, and biological features. The immune systems of mice and humans differ significantly, resulting in variances in the activity and behavior of their respective immunoglobulin subclasses. However, both human and murine IgG1 have a higher affinity for inhibitory Fc gamma receptor II b, and human and mouse Fc gamma receptor 1 are orthologs (https://www.scirp.org/journal/paperinformation?paperid=30589).

We modified our discussion in the revised version of the manuscript, lines 467-470, “In this study, IgG1 levels mainly increased following the immune challenge, like our previous study on the Staphylococcus challenge in mice [25]. IgG1 is vital for promoting opsonization and may have a remarkable ability to interact with FcγRs and activate the immune system.”

Comment: The sentence starting in line 417 seems incomplete.

Response: Thanks for pointing this out. We have now rewritten this sentence and included a reference.

Now, in Lines 587-594, “Previous studies that reported a decrease in IgG galactosylation under inflammatory conditions and with increasing age were reviewed by Gudelj et al [5]. In contrast, our findings demonstrated that young mice exhibit an increase in galactosylation upon immune challenge. This discrepancy may be attributed to immunological stimulation having a greater impact on sialylation in mice, while galactosylation is more commonly associated with humans. Furthermore, this was accompanied by a parallel reduction in IgG-Fc sialylation, a phenomenon previously observed in arthritis patients treated with anti-TNF [44].”

Comment: Lines 389, 398, 416, discussion about how the current study’s data are contradictory to the previous literature is not sufficient. More details about how it contradicts and potential reason why should be given.

Response: Thank you for this suggestion. We have now included more details, including references to potential reasons, in the revised version of the manuscript.

Now, in Lines 479-486, ”IgG3 has structural flexibility in the hinge region compared to other IgG isotypes, which makes it a potent antibody and highly efficient in mediating Fc-mediated immune functions [32]. A previous study in mice immunized with Legionella pneumophila demonstrated a significant decrease in the serum IgG3 response [33]. In the present study, we found that age, rather than immunological induction, predominantly increases IgG3 levels. Additional studies have suggested that IgG3, unlike IgG1, IgG2a, and IgG2b, but similar to IgM, enhances antibody responses through complement activation [34].”

Now, in Lines 491-496, ”Echoing our prior work with Staphylococcus-challenged in mice [25], this study found that sialic acid levels on total IgG increase with age. This elevation correlates with the sialylation of the Fab portion of IgG, which exhibits higher sialylation compared to the Fc portion. Furthermore, total IgG sialylation was assessed using Lectin ELISA, a method that detects sialic acid present on both the Fab and Fc parts.”

Now, in Lines 587-594, “Previous studies that reported a decrease in IgG galactosylation under inflammatory conditions and with increasing age were reviewed by Gudelj et al [5]. In contrast, our findings demonstrated that young mice exhibit an increase in galactosylation upon immune challenge. This discrepancy may be attributed to immunological stimulation having a greater impact on sialylation in mice, while galactosylation is more commonly associated with humans. Furthermore, this was accompanied by a parallel reduction in IgG-Fc sialylation, a phenomenon previously observed in arthritis patients treated with anti-TNF [44].”

Thank you for considering our work.

Sincerely,

Cecilia Engdahl PhD,

Associate Professor

Sahlgrenska Osteoporosis Centre, Centre for Bone and Arthritis Research

Institute of Medicine, Sahlgrenska Academy, University of Gothenburg

Gothenburg, Sweden

Reviewer 3 Report

Comments and Suggestions for Authors

In the manuscript, the authors have addressed the question regarding modifications to mouse IgG isotypes in young and aged mice, and the impact on the development of disease in a methylated-bovine serum albumen (mBSA)-induced mono-arthritis model.

The authors have done an excellent job in the development of the arthritis model, as presented in Figure 1 of the manuscript.  They further demonstrate that the total IgG generated in response to the mBSA following initial priming in complete FCA, followed by focused injection into the knee joint was not different between the young and adult mice.  The main finding of the study is the demonstration of different sialylation patterns of IgG2b and IgG3.  The finding is well presented and logically linked to pathology.

Specific Comments

1.     The authors have compared the responses in young (growing) mice of 2.5 months of age, and adult (non-growing) mice of 5.0 months of age.  While this is a meaningful comparison, and there are clear differences in the severity of mBSA-induced arthritis in the adult mice compared to the younger mice, it is not clear what was the justification for choosing these two age groups.  Clarification of this in the manuscript would be helpful.

2.     While the authors have demonstrated different sialylation patterns in two IgG isotypes associated with age to the mBSA-induced arthritis, there is no direct evidence that these sialylation patterns are specifically associated with increased pathology.  Direct evidence, through the infusion of “aged” serum into young mice and determining if the severity of disease is impacted would strengthen the argument.

3.     There are sections of the discussion that are not very clear in their presentation.  Minor editing of the discussion is required.

Comments on the Quality of English Language

Overall, excellent, with some minor editing required in the Discussion.

Author Response

Subject: Response to reviewer comments 

Manuscript ID: vaccines-2953621

Reviewer 3

Thank you for helping us improve our manuscript. We are delighted that you find our manuscript interesting, and we have revised our manuscript and for the constructive comments from you. All comments are responded to in the point-by-point response.

Comment: The authors have compared the responses in young (growing) mice of 2.5 months of age, and adult (non-growing) mice of 5.0 months of age.  While this is a meaningful comparison, and there are clear differences in the severity of mBSA-induced arthritis in the adult mice compared to the younger mice, it is not clear what was the justification for choosing these two age groups.  Clarification of this in the manuscript would be helpful.

Response: Thank you for your comment. The choice of the two age groups, young mice at 2.5 months and adult mice at 5.0 months, was driven by our aim to explore the scientific gap between young and adult mice in the context of mBSA-induced arthritis. These specific ages were selected to highlight any potential differences in disease progression and immune response due to age-related physiological changes that are pronounced between these stages. Additionally, our ethical permit restricts the use of older mice, which limited our ability to include more advanced age groups in this study.

Comment: While the authors have demonstrated different sialylation patterns in two IgG isotypes associated with age to the mBSA-induced arthritis, there is no direct evidence that these sialylation patterns are specifically associated with increased pathology.  Direct evidence, through the infusion of “aged” serum into young mice and determining if the severity of disease is impacted would strengthen the argument.

Response: Thank you for your constructive feedback regarding the linkage between the observed sialylation patterns and the pathology of mBSA-induced arthritis. We acknowledge that our current study does not provide direct causal evidence linking changes in sialylation with increased swelling over the joint. However, our study primarily aimed to illustrate the significance of different IgG subclass glycosylation in immune effector functions and to examine how subclass-specific glycosylation patterns change following immune activation, adulthood, or aging.

To clarify that we have no direct link between knee swelling and IgG glycosylation we rephrase the discussion Lines 446-454 ”First, we characterized the arthritis model and showed more pronounced knee swelling in adults compared to their younger counterparts. However, upon examining the arthritic joint, neither synovitis nor bone erosion responded differently in young mice compared to adult mice. This suggests that age could contribute to an increase of inflammatory mediators affecting fluid accumulation in the inflamed joints.”

Comment: There are sections of the discussion that are not very clear in their presentation. Minor editing of the discussion is required.

Response: Thanks for pointing this out. We have now further rephrased several parts of the discussions, including the following clarifications.

Now, in Lines 479-486, ”IgG3 has structural flexibility in the hinge region compared to other IgG isotypes, which makes it a potent antibody and highly efficient in mediating Fc-mediated immune functions [32]. A previous study in mice immunized with Legionella pneumophila demonstrated a significant decrease in the serum IgG3 response [33]. In the present study, we found that age, rather than immunological induction, predominantly increases IgG3 levels. Additional studies have suggested that IgG3, unlike IgG1, IgG2a, and IgG2b, but similar to IgM, enhances antibody responses through complement activation [34].”

Now, in Lines 491-496, ”Echoing our prior work with Staphylococcus-challenged in mice [25], this study found that sialic acid levels on total IgG increase with age. This elevation correlates with the sialylation of the Fab portion of IgG, which exhibits higher sialylation compared to the Fc portion. Furthermore, total IgG sialylation was assessed using Lectin ELISA, a method that detects sialic acid present on both the Fab and Fc parts.”

Now, in Lines 587-594, “Previous studies that reported a decrease in IgG galactosylation under inflammatory conditions and with increasing age were reviewed by Gudelj et al [5]. In contrast, our findings demonstrated that young mice exhibit an increase in galactosylation upon immune challenge. This discrepancy may be attributed to immunological stimulation having a greater impact on sialylation in mice, while galactosylation is more commonly associated with humans. Furthermore, this was accompanied by a parallel reduction in IgG-Fc sialylation, a phenomenon previously observed in arthritis patients treated with anti-TNF [44].”

Thank you for considering our work.

Sincerely,

Cecilia Engdahl PhD,

Associate Professor

Sahlgrenska Osteoporosis Centre, Centre for Bone and Arthritis Research

Institute of Medicine, Sahlgrenska Academy, University of Gothenburg

Gothenburg, Sweden

Round 2

Reviewer 2 Report

Comments and Suggestions for Authors

The authors have modified the manuscript in response to the recommendations, which has overall improved the manuscript. I have no more comments.